# Nephrite-Bearing Mining Waste As a Promising Mineral Additive in the Production of New Cement Types

**Liudmila I. Khudyakova [1], Evgeniy V. Kislov [2,\*], Pavel L. Paleev [1] and Irina Yu. Kotova [1]**

[1] Baikal Institute of Nature Management, Siberian Branch of the Russian Academy of Sciences (BINM SB RAS), 6, Sakhyanovoy str., Ulan-Ude 670047, Russia; lkhud@binm.ru (L.I.K.); palpavel@mail.ru (P.L.P.); ikotova@binm.ru (I.Y.K.)

[2] Geological Institute, Siberian Branch of the Russian Academy of Sciences (GIN SB RAS), 6a, Sakhyanovoy str., Ulan-Ude 670047, Russia

\* Correspondence: evg-kislov@yandex.ru

**Abstract:** A growing demand for products made of jewelry and ornamental stones, including nephrite, requires an increase in mining volume. However, only less than 30% of the extracted raw material is suitable for processing. The rest of the low grade nephrites are substandard and unclaimed, and they negatively affect various life spheres. In this regard, their involvement in industrial turnover is an actual task. One of the directions of mining waste use is production of building materials, in particular, cements. The low grade nephrite can act here as mineral additives. In the course of the research, the optimal amount of low grade nephrite waste additive was determined, which is 30% of the cement mass. The grinding time of a raw mix is 10 min. It was found that introduction of the additive affects the hydration activity of cement compositions. Compressive strength of the mixed cement is 25% higher than that of the control sample. At the same time, new phases in the hydrated cement were not recorded. Good physical and mechanical properties of the obtained cements are achieved when hardening in normal humidity conditions. Heat and humidity treatments do not facilitate the hydration processes in binary systems. The conducted studies have shown that low grade nephrite can be used as mineral additives in cement production. This will allow development of not only a new type of product, but also reduction of the negative impact of cement production on the environment.

**Keywords:** nephrite deposits; low-grade nephrite; mineral additive; cement

## 1. Introduction

Building materials are commercial products consisting almost entirely of non-renewable mineral resources [1,2]. These include Portland cement, which is widely used in building [3–5]. In addition to the fact that its production consumes significant amounts of mineral raw materials, it is among the leaders in polluting the environment with carbon dioxide emissions [6–9]. To minimize the negative impact, the amount of clinker in the composition of the cements is reduced and replaced with mineral additives of various types [10].

Mining waste is mainly used as mineral additives, including quartz, marble, basalt dust, granite, limestone powders, fly ash, etc. Moreover, not just one type of additive is used, but very often several types simultaneously. The chemical and mineralogical composition of cements plays an important role in the kinetics of hydration of the cements with mineral additives [11].

It is established that use of quartz dust in the cement composition increases mechanical strength. Combined presence of colemanite with the waste leads to an increase in specific surface area and

water consumption, as well as a decrease in strength properties [12]. Addition of dolomite to the cement composition reduces its water demand, without having a positive effect on the mechanical characteristics. On the contrary, addition of coal industry waste increases microporosity of the materials and boosts their strength [13]. Replacement of 20% of Portland cement with activated coal mining wastes (kaolinite) slightly reduces mechanical properties of the obtained materials, which, however, are within the requirements of technical documentation [14,15]. Besides, coal combustion waste has pozzolanic activity [16]. Zeolites and bentonites can be added to cement binders to bind the heavy metals in the fly ash [17]. Compositions with zeolite manifest themselves best, since their mechanical characteristics have increased values [18].

The dust generated during the development of mineral deposits has prospective use as a mineral additive. Marble quarries produce a large amount of marble dust, which can be used as an additive in cements. At 5% of its replacement, the best physical and mechanical properties are observed, and they correspond to an improved cement grade [19]. Limestone dust in the amount of up to 8 mass% used in production of cement pastes and solutions has a noticeable effect on their properties. The setting time of the samples is extended, their density increases, and the mechanical properties improve [20]. Chalcedony dust, introduced in an amount of 25%, determines gains of strength of the samples at the later time of hardening [21].

During gold mining, huge volumes of gold-bearing waste remain in the quarries. They consist of quartz, muscovite, albite, and microcline. Their introduction into the composition of the feed in the production of cements improves the hydration process and increases the strength of commercial products [22,23]. Addition of up to 15% heat-treated heavy loams with a specific surface area not exceeding 500 $m^2$/kg leads to an increase in density, water resistance, and strength of cement compositions [24].

Use of magnesium-containing additives in the composition of mixed cements is of interest. Introduction of their optimal amount leads to an increase in strength of the obtained compositions. For basalts, this parameter is up to 12 mass%. Water demand remains at the same level, and process of setting the cement paste slows down [20,25–27]. Increasing the amount of the additive leads to a decrease in strength properties at the early stages of hydration (7 and 28 days). However, at later age (91 days), the situation is reversed: a mixed cement paste has a higher strength than a control sample [28].

In addition, basalt crushing waste can be used as raw materials in the production of Portland cement clinker. The obtained clinkers do not differ in their phase composition from non-additive ones, and the properties of Portland cements meet the requirements of the technical conditions [29]. Basalt powder added to the composition of solutions does not have a positive effect on their quality. It acts as inert filling materials [30].

In preparation of binding compositions, it is possible to use mining waste containing magnesium hydrosilicates, such as talc or serpentinite. Mixing of talc with a solution of sodium polyphosphate in the combined presence of magnesia makes it possible to obtain high-strength materials [31]. Adding the serpentinite to the composition of magnesia cement increases its durability [32]. Its application as an additive to Portland cement clinker improves the mechanical properties [33]. In addition to the mentioned above rocks, dunites, wehrlites, and troctolites are promising raw materials for cement production [34]. Using them as a mineral additive improves the exploitative characteristics of the obtained materials.

The introduced additives affect the hydration capacity of cement compositions in various ways. Fly ash, as well as zeolite and bentonite, do not lead to the appearance of new phases in the hydrated cement, but only accelerate the beginning of the hydration process and its complete finishing [17]. When 20% of activated clay materials (kaolinite) are added to the mixed cement, an increase of the concentration of alumina phases $C_4AH_{13}$, $C_4AcH_{12}$, and polymerized C–S–H gel is observed [15].

Dolomite and coal waste facilitate the formation of $C_4AcH_{12}$ crystal phase together with ettringite and C–S–H gel. Moreover, in the latter case its amount is higher, which facilitates the compaction of

cements [13]. Introduction of gold-bearing tailings as silica raw materials also does not lead to formation of new phases during the hydration of cements; however, it boosts the reactivity. The products of hydration are ettringite, calcium hydroxide, and the gel-phase C–S–H [22]. The applied limestone quarry dust acts as filler, reacting with the products of cement hydration to form calcium bicarbonate. Basalt quarry dust is not only filler, but also a pozzolanic additive. When hydrated in a mixed cement composition, it leads to the formation of an additional amount of calcium hydrosilicates C–S–H, which determines mechanical strength of the material [20]. In addition, the interaction rate of silica, which is a part of basalts, with $Ca(OH)_2$, formed as a result of the cement hydration increases [28]. Chalcedony dust also reacts with portlandite to form calcium hydrosilicates [21].

Thus, the search for new types of mineral additives and the study of their influence on the hydration activity of Portland cement is an urgent task.

## 2. Materials and Methods

### 2.1. Materials

One of the mineral additives that affect the properties of cement binding materials is nephrite-bearing waste formed as a result of extraction and isolation of high-grade nephrite. Their number exceeds 70%. Being among the unclaimed ones, the waste negatively affects the environment.

Nephrite is a rock, the bulk of which is tremolite with an isomorphic admixture of iron. Its peculiarity is high viscosity determined by the interwoven-fibrous microstructure. Nephrite is characterized by textural uniformity and various colors: green of different shades, white, light yellow, and rarely brown-red and black. The color distribution is usually uneven: cloudy, striped, and spotted. Even solid color is less common and most valued. The stone is translucent to a different depth. With a dense uniform texture, it has good polish ability with greasy to matte luster, an uneven, hackly (shistic variation), or conchoidal fracture. Hardness of nephrite is 6.0–6.5 (Mohs scale), for nephrite containing talc and serpentine it is up to 5.5. The main properties that determine the quality of nephrite and affect its price are color, translucency, ability to accept polishing, and size of continuous sections [35].

Nephrite is a highly precious jewelry and ornamental stone that has long been used by humanity, and is particularly popular in China and some other countries. The most valued types of nephrite are translucent white and bluish-green with a minimum amount of ore minerals. It is rare to find a valuable jewelry variety of nephrite with the effect of a cat's eye. Nephrite is considered one of the best materials for stone-cutting (vases, cups, caskets) and jewelry (inserts in jewelry, rings, signets, entire bracelets) products. It is also used as a decorative and finishing material for interior decoration (production of mosaic panels, tabletops) [35].

According to Harlow and Sorenson [36], nephrites can be grouped into two genetic types: (1) serpentinite type and (2) marble type. The serpentinite type nephrite forms by replacement of serpentinite by Ca and Si. The marble type forms by metasomatic exchange of dolomite with Si-saturated, $H_2O$-rich fluids that are commonly associated with granitic plutons [36]. Exogenous placers, usually alluvial, are associated with the root sources [37,38]. Deposits of the serpentinite type are the source of green nephrite, changed to brown by surface processes, and black nephrite is rare. There are the Academicheskoe and Nyrdomenshor deposits at the Ural, the Kurtushibinskoe and Kantegirskoe deposits at the Western Sayan, Ospinskoe, Gorlykgolskoe, Ulankhodinskoe, and other deposits at the Eastern Sayan, Khangarulskoe, Khamarkhudinskoe, and the Khargantinskoe deposits at the Western Khamar–Daban, and the Paramskoe deposit at the Middle Vitim area of Russia. Deposits of the marble type are the source of light-colored nephrite from white to light-green, and brown and dark, dendritic patterns are rare. They are the Kavoktinskoe, Khaytinskoe, Golyubinskoe, Buromskoe, Udokanskoe, Ollomi, and other deposits at the Middle Vitim area in Russia [39].

As the research objects, nephrite-bearing rocks of the Ulankhodinskoye deposit of the Eastern Sayan were used.

The Ulankhodinskoye (Ulan–Khodynskoe, Kharanurskoye) deposit was discovered in 1965, and it is situated within the Kharanursky (Kholbyn–Khairkhansky) ultramafite massif at the southeastern part of the Eastern Sayan, Okinskiy district of the Republic of Buryatia (Russia) near the border with Mongolia.

The Kharanursky massif is located in the upper reaches of the Urik river basin. The shape of the massif is pear-like and irregular. Its maximum width is 6.6 km, its length is 12.2 km, and its area is about 25 km$^2$. The contacts of the massif along with the host Proterozoic rocks are tectonic [40].

In the central part of the Kharanursky massif, which is associated with the deposit, on an area of about 6 km$^2$ cataclasites after chrysotile serpentinites are common. Carbonated serpentinites and talc–carbonate rocks are less common [40]. Ultrabasic rocks in nephrite-bearing zones are intruded by small bodies of gabbro-diabases, plagiogranites, and plagioporphyris, which are associated with nephrite formation. Within the massif, except the known depleted Kharazhalginskaya nephrite vein, the Ulankhodinskoye deposit which is unique in the quality and light green color of nephrite was discovered [41].

In 1965–1966, 11 veins were opened here. To date, their number has reached 21, grouped in 2 nephrite-bearing zones. In the nephrite-bearing zones, meta-dikes of basic and acidic composition are developed. Metasomatic processes have led to a significant change in the primary composition of dikes. In a number of cases, metasomatic processes in the contact zones of dikes determined the formation of nephrite veins after serpentinites [40,41].

Veins 9 and 10 are unique in quality and lie en echelon at the salbands of the underlying and overlying exocontacts of the dike of amphibolized gabbro changed to zoisite–diopside–quartz rock (Figure 1). Veins 9 and 10 have the north-western extension (295–300°), and their dip is at an angle of 50° to the south-west. The vein length is 9.0 and 9.5 m, respectively, and the average width is 1.2 and 1.3 m. To the depth of 10 m, the veins were studied by drilling wells and a quarry. The marginal parts of the veins are composed of nephrite shist of brownish-green color, while in the central part of the vein the nephrite is light green and massive. Currently, the veins are mostly depleted [41].

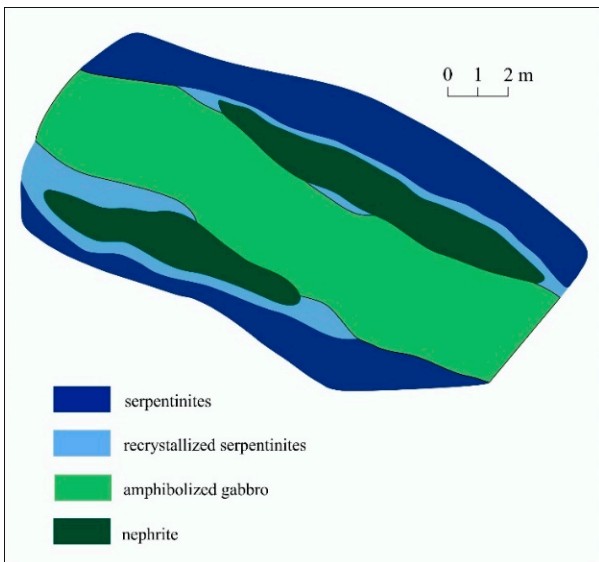

**Figure 1.** Scheme of the geological position of the nephrite veins of the Ulankhodinskoye deposit according to Suturin and Zamaletdinov [42] with modifications.

The nephrite quality of the Ulankhodinskoye deposit is high—a pleasant light green color with a minimum number of inclusions of extraneous minerals and rocks. In the polished samples, light green nephrite is translucent in plates more than 20 mm thick [43]. The crystal aggregates of tremolite composing the nephrites are a bundle of one to two dozen curved or parallel tremolite fibers that form petal-shaped aggregates that are differently oriented relative to each other. The ends of the fibers are

intertwoven, and the thickness of the fibers is approximately the same (0.01–0.02 mm). Of the accessory minerals in nephrite, there are uniformly dispersed rounded grains of chromite up to 1 mm in size [40].

A significant part of the nephrite is low grade—brown-green nephrite with a well-developed schistosity of the contact parts of the veins, areas of cracks and zones of cataclase, talc, and tremolitization. Replacement of light-green nephrite with brownish-green occurs near cracks and in areas of the vein that are exposed to microcatalase. In areas where the catalase process reaches its maximum size, nephrite recrystallization occurs, while the nephrite loses its green color and becomes grayish without ore inclusions. Brown-green nephrite retains a similar microstructure, texture, and ore minerals inclusions. The size of petal-shaped tremolite aggregates in them is from $0.05 \times 0.06$ to $0.8 \times 0.9$ mm. In polished samples, brown-green nephrite is translucent to a depth of 10 mm [41].

### 2.1.1. Low Grade Nephrite of the Ulankhodinskoye Deposit

The chemical composition of brownish-green nephrite schist is shown in Table 1.

**Table 1.** Chemical composition of low grade nephrite, mass%.

| Basic Oxides | | | | | | | | |
|---|---|---|---|---|---|---|---|---|
| $SiO_2$ | $Al_2O_3$ | $Fe_2O_3$ | FeO | MgO | CaO | $Na_2O$ | $K_2O$ | LOI |
| 56.20 | 0.94 | 0.21 | 3.28 | 22.37 | 13.48 | 0.04 | 0.03 | 3.41 |

Note: This and the following analyzes were performed by atomic absorption, spectrophotometric, flame photometric, gravimetric, titrimetric methods at the Analytical Center of mineralogical, geochemical and isotope studies at the Geological Institute, SB RAS Ulan-Ude, Russia, analysts A.A. Tsyrenova, T.I. Kazantseva, V.A. Ivanova.

It is established that the studied rocks are low-iron tremolites, the main content of which is represented by $SiO_2$, MgO, CaO, and FeO. The results of X-ray phase analysis show that the rocks consist of tremolite–ferroactinolite minerals; the sample corresponds to the minerals tremolite (t) $Ca_2Mg_5[Si_4O_{11}]_2(OH)_2$ and actinolite (a) $Ca_2(Mg,Fe)_5[Si_4O_{11}]_2(OH)_2$ (Figure 2).

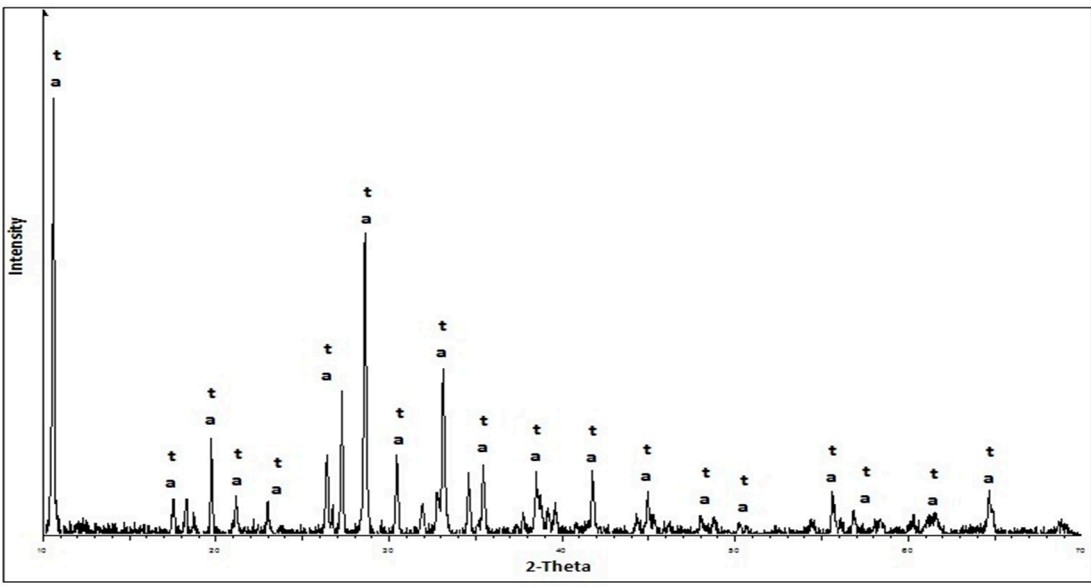

**Figure 2.** X-ray phase analysis of a low grade nephrite.

Physical mechanical and chemical properties of the low grade nephrite were determined. These are high tensile rocks with a grade of crushing capacity 1600 and abrasion I. They have high fracture strength, are hard to be broken into pieces, but are ideal for carving. The low grade nephrite is resistant to acids and all types of decay, and it is not exposed to the environment. The true density of the rocks is 3094 kg·m$^{-3}$. The following was identified: silicate (silica) module—12.66; acidity module—1.59;

basicity module (hydraulic module)—0.63; quality coefficient (hydraulic activity)—0.66. The rock belongs to acidic and active types. It does not harden on its own, but can be used as a mineral additive in the cement composition.

### 2.1.2. Portland Cement Clinker

To obtain composite binding materials, Portland cement clinker of Timluy cement plant (LLC "Timlyuytsement", Buryatia, Russia) was used. The clinker mixture consists of tricalcium silicate $3CaO–SiO_2$ ($C_3S$), bicalcium silicate $2CaO–SiO_2$ ($C_2S$), tricalcium aluminate $3CaO–A1_2O_3$ ($C_3A$), and four-calcium aluminoferrite $4CaO–Al_2O_3–Fe_2O_3$ ($C_4AF$), the content of which is shown in Table 2. The chemical composition of the clinker is shown in Table 3.

**Table 2.** Contents of basic minerals, mass%.

| Basic Minerals | | | | |
|---|---|---|---|---|
| $C_3S$ | $C_2S$ | $C_3A$ | $C_4AF$ | Different |
| 60.0 | 17.0 | 6.0 | 13.0 | 4.0 |

**Table 3.** Chemical composition of the Portland cement clinker, mass%.

| Basic Oxides | | | | | | | |
|---|---|---|---|---|---|---|---|
| $SiO_2$ | $Al_2O_3$ | MgO | CaO | $Fe_2O_3$ | $Na_2O + K_2O$ | $SO_3$ | LOI |
| 21.60 | 4.92 | 0.91 | 64.88 | 4.31 | 1.17 | 0.89 | 1.04 |

The Portland cement clinker meets the requirements of the Interstate standard GOST 31108–2016 [44] "Common cements. Specifications" (Russia).

### 2.1.3. Additional Materials

In the course of the research, two-water gypsum of Nukutsky gypsum quarry (Russia, Irkutsk region) was used, and its chemical composition is shown in Table 4.

**Table 4.** Chemical composition of two-water gypsum, mass%.

| Basic Oxides | | | | | | | | |
|---|---|---|---|---|---|---|---|---|
| $SiO_2$ | $Al_2O_3$ | $Fe_2O_3$ | MgO | CaO | $Na_2O$ | $K_2O$ | $SO_3$ | LOI |
| 1.76 | 0.03 | 0.07 | 4.27 | 31.17 | 0.07 | 0.03 | 40.92 | 17.25 |

The used gypsum $CaSO_4·2H_2O$ meets the requirements of the Interstate standard GOST 4013–82 [45] "Gypsum and gypsum anhydrite rock for the manufacture of binders. Specifications" (Russia).

To close binding compositions tap water was used, and the water meets the requirements of the Interstate standard GOST 23732–2011 [46] "Water for concrete and mortars. Specifications" (Russia).

### 2.2. Methods

### 2.2.1. Sample Preparation of Cement Binding Compositions

For the research, mixtures of Portland cement clinker with the addition of nephrite-bearing rocks in an amount of 10–40% were prepared. The mixtures represented the following proportions of the clinker and the rock: 90:10; 80:20; 70:30; 60:40. Gypsum dihydrate was added to each one—3% of the mass of the mixture. The size of the raw material pieces was less than 100 mm. The prepared feed was closed with water at a water-solid ratio of 0.3. Samples of cement dough were formed—cubes of 20 × 20 × 20 mm in size from a solution of normal density controlled by a Vic device.

After 24 h of hardening in normal humidity conditions, the samples were removed from the mold. One part was placed in a bath with a hydraulic shutter with water temperature of (20 ± 1) °C. Another part of the samples was exposed to heat-and-water treatment (HMT) by steaming in a laboratory steaming chamber with automatic temperature control. Treatment mode: temperature increase to 100 °C—2 h; isothermal heating in boiling water vapor—5 h; cooling—2 h.

The method of forming samples, conditions, and terms of hardening meet the national standards of the Russian Federation. The samples were tested after 7, 14, and 28 days of hardening, as well as after heat and humidity treatment.

All the tests were performed in three repetitions. The properties in the article are the average value of the data obtained.

### 2.2.2. Methods of the Research of Cement Compositions

The research methodology included performing chemical, X-ray, and phase analyses, as well as physical and mechanical tests.

Chemical analysis was performed by methods of atomic absorption spectroscopy using a Unicam spectrophotometer SOLAAR–6M (Thermo Electron, Franklin, Ma, USA) with the suitable software and gravimetry using an electronic scale VSL–200/0,1A (Nevskiye Vesy, St. Petersburg, Russia).

Titrimetric analysis was used to determine the pozzolanic activity of the low grade nephrite.

XRD analysis was performed using an X-ray diffractometer (Bruker AXS, D8–Advance, Bruker, Karlsruhe, Germany) with a Cu tube and a scanning range from 10 to 70 2theta, with a step of 0.02 and 0.2 s/step—1 measuring time.

Mechanical tests were performed on a PG–100 test hydraulic press (DEG, St. Petersburg, Russia) with a load range of up to 10 t and a plate movement speed of 10 ± 1 mm/min.

## 3. Results and Discussion

The research included determination of pozzolanic activity of the low grade nephrite, which is used as a mineral additive in cements. This type of study is usually conducted using the traditional method based on absorption of lime from lime solution. However, in the work [47] it was found that the accelerated Chapel method is preferable [48], which allows us to effectively determine activity of additives of different quality.

Pozzolanic activity of the low grade nephrite determined by the Chapel method was 66 mg $g^{-1}$. It indicates that this raw material is not inert and can be used in the production of cements with mineral additives.

The influence of the amount of the additive of the low grade nephrite and the time of grinding of the raw material mixture on mechanical properties of composite binders was studied. In addition, the dependence of their strength on the hardening conditions was established.

As can be seen from the data obtained, mechanical activation of the raw material mixture has a significant impact on the strength of cement compositions (Figure 3).

The amount of the additive of nephrite-bearing waste in the cement compositions also affects their hydration activity (Figure 4). The lowest strength properties were observed in samples with 10% of the rock additive, the highest with 30% of the additive. When 40% of the waste was added to the raw material feed, the strength properties of the hydrated samples were practically at the same level and did not depend on the duration of grinding.

It is established that the strength set of binding compositions depends on the conditions of their hardening. From the data in Figure 5, presented for samples crushed within 10 min, it is clear that for all types of compositions, heat-and-water treatment did not lead to an increase in the hydration activity of cement compositions. The best results were obtained when hardening in normal humidity conditions. The main strength gain occurred during the first 7 days of sample hardening (more than 70% of the 28-day strength). By fourteen days, this property was more than 90% of the 28-day strength.

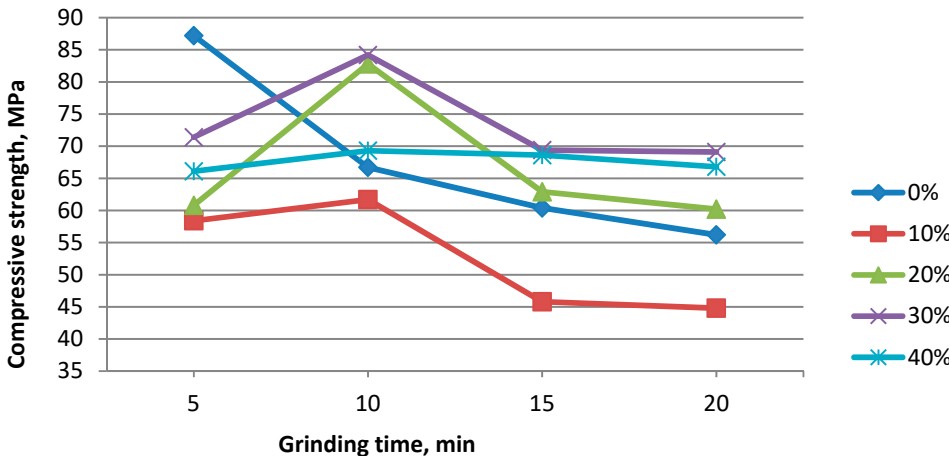

**Figure 3.** Dependence of the strength of composite binders on the time of grinding of the raw mixture and the amount of additives of substandard nephrite.

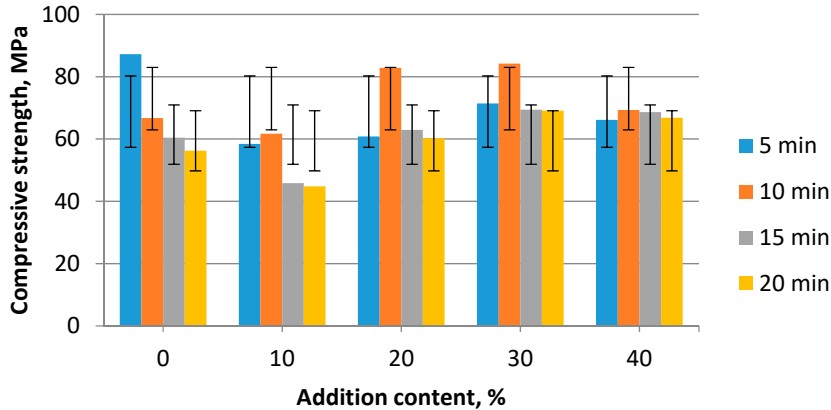

**Figure 4.** Dependence of the strength of composite binders on the amount of additives of the substandard nephrite.

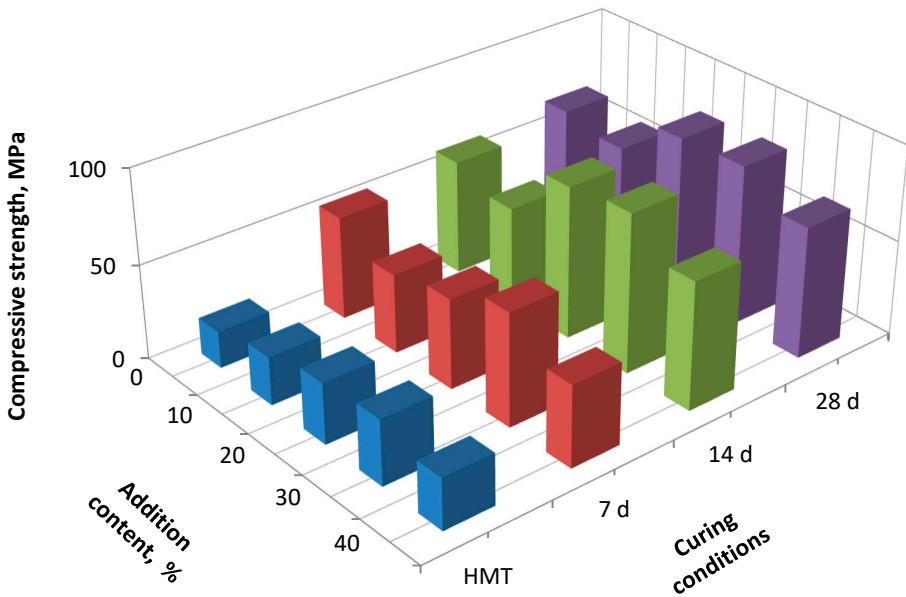

**Figure 5.** Dependence of the strength of the binders with different amounts of additives on the hardening conditions.

X-ray phase analysis of samples after 28 days of hardening in normal humidity conditions is shown in Figure 6.

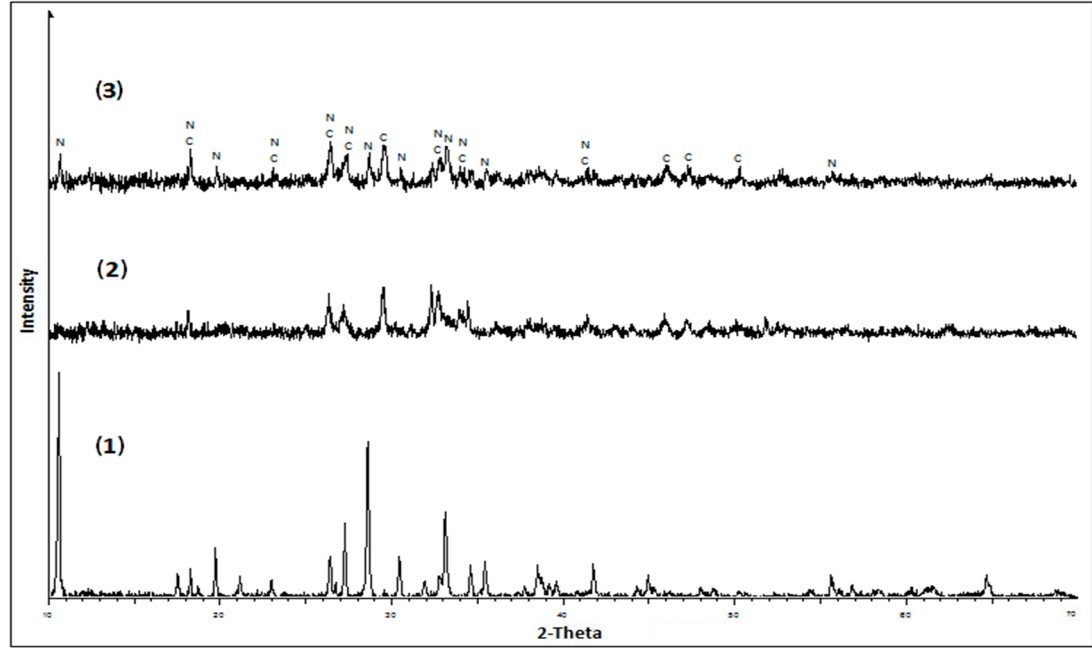

**Figure 6.** X-ray phase analysis of the cement stone after 28 days of hardening: (1) substandard nephrite (n); (2) Portland cement (c); (3) cement with a 30% nephrite additive.

An X-ray diffractogram of a binder composition with a 30% mineral additive (X-ray 3) is a combination of minerals that is a direct result of hydration of pure cement (X-ray 2) and its subsequent hardening. Most of the X-ray reflections were located between 25–35 ($2\theta$), which mainly correspond to the phases of Portland cement. In addition, the X-ray showed reflexes related to nephrite minerals (X-ray 1).

In the process of hydration of Portland cement with a mineral additive, the appearance of reflexes of new phases was not observed. Binary cement compositions containing substandard nephrite behaved in the same way as standard Portland cement. The addition of nephrite does not interfere with the process of its hydration.

As a result of the work, new types of cements with the addition of low grade nephrite were obtained and their physical and mechanical properties were studied (Table 5).

**Table 5.** Physical and mechanical properties of cements.

| Properties | Cement with Nephrite Additive | Portland Cement 400 D0 |
| --- | --- | --- |
| Start of setting | 4 h 51 min | 3 h 20 min |
| End of setting | 7 h 45 min | 5 h 20 min |
| Dissolving of cone | 117 | 114 |
| Compression strength, MPa | 84.2 | 66.7 |
| Compression strength after HMT, MPa | 34.4 | 19.4 |
| Mean density, kg/m$^3$ | 2256 | 2315 |

According to the presented data, it can be seen that cements with the addition of low grade nephrite differ in their physical and mechanical properties from Portland cement. It was found that the compressive strength of the mixed cement was higher than that of the control sample and was 84.2 MPa. Unlike Portland cement, the duration of hardening of which is within 2 h, this process of

the obtained material takes 2 h and 54 min. The average density of the samples with the addition of nephrite was lower than that of Portland cement, and was equal to 2256 kg/m$^3$.

The index of pozzolanic activity of the low grade nephrite for compressive strength of cement–sand samples at the age of 28 days of normal-humidity hardening was determined. Depending on the amount of rock admixture in the composition of cements, the activity index (in %) was equal to: at 10–93.6%; at 20–99.1%; at 30–103.5%; at 40–95.0%. The obtained data indicate that addition of the low nephrite into the cement compositions in the amount of 20–30% did not significantly affect the strength of the cement–sand compositions, which were within the strength of the control sample.

Use of the nephrite has some environmental aspects. That is due to a fact that nephrite is composed of tremolite which during milling/grinding generates a highly carcinogenic tremolite dust similar to asbestos. Nephrite dust increases a risk of a lung cancer as well as other types of cancer. So, if tremolite has to be milled during a proposed technological process, then the entire industry needs to be located far from inherited areas and workers need to wear safety equipment (at least masks preventing from highly toxic dusts).

## 4. Conclusions

As a result of the conducted studies, it was found that low grade nephrite of the Ulankhodinskoye deposit of the Kharanur massif of the Eastern Sayan can be used as a mineral additive in cement production. It was determined that the studied rocks belong to low-iron tremolites and consist of minerals of the tremolite–ferroactinolite series.

Composite binding materials with the addition of low-grade nephrite were obtained. It is established that their physical and mechanical characteristics depend on the amount of additives, the time of grinding the raw mixture, and the conditions of hardening.

It was found that the introduction of a mineral additive in the form of low grade nephrite does not lead to the appearance of new phases in the hydrated cement, i.e., it does not interfere with the process of its hydration. However, the strength properties of mixed cement increase. Thus, the compressive strength of the obtained sample is 25% higher than that of the standard sample, and is 84.2 MPa.

It should be noted that, in contrast to normal-humidity hardening, heat-and-water treatment does not lead to an increase in the hydration activity of cement compositions. It was determined that the optimal technological parameters for obtaining new types of materials are: the grinding time of the raw mixture—10 min, the amount of additives—30%, and the hardening conditions—28 days of normal humidity hardening.

The physical and mechanical properties of the obtained materials meet the requirements of the national standard of the Russian Federation.

Thus, waste from the extraction of varietal nephrite can be used as a mineral additive in the production of mixed cements. Their involvement in industrial turnover will help to reduce the negative impact of the mining industry and cement production on the environment.

**Author Contributions:** L.I.K. conceived the idea, designed the experiments, analyzed the results, and reviewed the paper. E.V.K. explored the deposit, contributed raw materials, and reviewed the paper. P.L.P. performed the experiments, analyzed the results, and reviewed the paper. I.Y.K. performed the analysis, analyzed the results, and reviewed the paper. All authors have read and agreed to the published version of the manuscript.

**Funding:** The work was carried out within the framework of state tasks of the GIN SB RAS, No. state reg. AAAA-A17-117011650012-7 and BINM SB RAS, No. state reg. AAAA-A17-117021310253-8.

**Acknowledgments:** The authors are grateful to Adam Abersteiner for helping to refine the English manuscript. The authors wish to express their sincere thanks to the journal editors and two anonymous reviewers for their constructive comments, which significantly improved the quality of the paper. The study was conducted using facilities of the Analytical Center of mineralogical, geochemical and isotope studies at the Geological Institute, SB RAS Ulan-Ude, Russia.

**Conflicts of Interest:** The authors declare no conflicts of interest.

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
