# Peer review of "Nephrite-Bearing Mining Waste As a Promising Mineral Additive in the Production of New Cement Types"

_minerals, doi:10.3390/min10050394_

Round 1
Reviewer 1 Report
Dear Editors,
It is a pleasure for me to review the manuscript entitled "Nephrite-bearing Mining Waste As a Promising Mineral Additive in the Production of New Cement Types". Your manuscript presents an interesting study concerning the usage of wastes, generated during the nephrite mining, as an additive to production of the cement. The proposed form of the nephrite mining wastes recycling tends to be promising in both the removal of mining wastes and enhancing the properties of the manufactured cement. Thus I strongly recomment the publication of the manuscript. However, a major improvement of the manuscript is desirable before publication.
For first, an environmental aspects of the study definitely need to be improved. I mean, the mineralogical-petrological composition of the nephrite mining wastes have to be described more detailed. Especially it should be clearly stated, whether a poor quality nephrite (e.g., nephrite schist) or wall rocks (serpentinite, metagranite, metabasite etc.), or both, belongs to this group. That is due to a fact, that nephrite is composed of tremolite which during milling/grinding generates a highly cancirogenic tremolite asbestos dust. Nephrite dust increases a risk of a lung cancer (Yang et al., 2013, Occupational and Environmental Medicine, 70) as well as other types of cancer (Yang et al., 2016, Occupational and Environmental Medicine, 73). So, if tremolite has to be milled during a proposed technological process, then the entire industry needs to be located far from inherited areas and workers need to wear safety equipments (at least masks preventing from highly toxic dusts). Similarly, numerous metabasites or serpentinites contain tremolite asbestos or chrysotile asbestos. All these informations certainly needs to be included in the first subchapters (e.g., in the introduction).
Moreover, in the introduction, also the nephrite genetic subdivision and formation mechaninsm need to be presented more precisely, because in their present form, these are far from being clearly presented according to the recent knowledge. Just two or three, but carefoully written sentences. Please refer to the one of following works for details: Harlow & Sorensen (2005, International Geology Review, 47), Harlow et al. (2007, Mineralogical Association of Canada Shourtcourse, 37) or Harlow et al. (2014, Mineralogical Association of Canada Shortcourse, 44). Also nephrite is composed of tremolite-ferroactinolite amphiboles - see the recent amphiboles classification scheme of Hawthorne et al. (2012, American Mineralogist, 97). Finally, it cannot be stated that dolomite marbles host only light colored nephrites as opposite to serpentinites. There are exceptions and some dolomite-hosted nephrites are even darker green than typical serpentinized peridotite-hosted nephrites, for example a ZĹ‚oty Stok deposit in Poland (Gil et al., 2015, Canadian Mineralogist, 53).
Moreover, I made some minor corrections directly to the text in the annoted file, which is an integral part of the review. However, the language check by a native speaker, before final publication, is still desirable.
Summing up, the manuscript have to be published in Minerals Journal, although implementation of the mentioned changes will make it stronger and of a higher impact, which will be profitable for both the Authors and Journal.
Sincerely,
The Reviewer

Reviewer 2 Report
This article is interesting but there is a huge lack of data to be able to confirm whether this material to be a major product for the substitution of cement.
-We do not have details on the specific surface used of nephrite, on its pozzolanic capacity, as well as on its microstructure. SEM tests would have been interesting?
-We don't have data on the fineness also of the cement? Or can there be a more or less favorable interaction with respect to the fineness of cements and additions?
-Please include standard deviations in the graphics.
-Why not observe by DRX the evolution of the quantity of hydrates formed over time? This confirms that the addition to a pozzolanic reaction?
-Please determine the pozzolanic activity index for nephrites?
- Why don't we have information on the properties in the fresh state of the material (heat generation, setting time, workability, etc.)?
Author Response
This article is interesting but there is a huge lack of data to be able to confirm whether this material to be a major product for the substitution of cement.
-We do not have details on the specific surface used of nephrite, on its pozzolanic capacity, as well as on its microstructure. SEM tests would have been interesting?
Current studies suggest the use of low grade nephrite as a mineral additive at the stage of cement production. Nephrite is added to Portland cement clinker and gypsum directly to the mill. The size of the nephrite corresponds to the size of Portland cement clinker. Everything is crushed to a certain specific surface area.
Changed – ll. 229, 248, 255-262
-We don't have data on the fineness also of the cement? Or can there be a more or less favorable interaction with respect to the fineness of cements and additions?
As a comparison, we used Portland cement, obtained by grinding Portland cement clinker with gypsum, with a specific surface area identical to cement with the addition of nephrite.
-Please include standard deviations in the graphics.
It is not possible to include standard deviations in the graph presented in the article, because it is made in the form of a volume histogram. Therefore, we added another graph to the article (Figure 4) with standard deviations.
Changed – ll. 272, 276-278
-Why not observe by DRX the evolution of the quantity of hydrates formed over time? This confirms that the addition to a pozzolanic reaction?
X-ray phase analysis data are presented for samples after 28 days of hardening under normal humidity conditions. The authors wanted to show that in the process of hydration of Portland cement with a mineral additive, the appearance of reflexes of new phases is not observed. Cement compositions containing nephrite are hydrated in the same way as Portland cement.
The dependence of cement hydration with the addition of jade on the time of hydration is not presented in the article.
-Please determine the pozzolanic activity index for nephrites?
Changed – ll. 313-318.
- Why don't we have information on the properties in the fresh state of the material (heat generation, setting time, workability, etc.)?
Changed – ll. 195-201.

Round 2
Reviewer 2 Report
Could you give more detail to figure 6 because it is not sufficiently explicit?
Author Response
Dear colleagues!
All points and responses are at the attached file
Thank you very mutch
Sincerely yours,
Evgeniy
